# Potential Role of Hydroxyapatite Nanocrystalline for Early Diagnostics of Ovarian Cancer

**DOI:** 10.3390/diagnostics11101741

**Published:** 2021-09-22

**Authors:** Ruslana Chyzhma, Artem Piddubnyi, Sergey Danilchenko, Olha Kravtsova, Roman Moskalenko

**Affiliations:** 1Department of Pathology, Sumy State University, 40007 Sumy, Ukraine; r.chyzhma97@gmail.com (R.C.); a.piddubny@med.sumdu.edu.ua (A.P.); eriugen@i.ua (O.K.); 2Ukrainian-Swedish Research Center SUMEYA, Sumy State University, 40022 Sumy, Ukraine; 3Department of Medical Biochemistry and Biophysics, Umea University, SE-90736 Umea, Sweden; 4Institute of Applied Physics NAS of Ukraine, 40007 Sumy, Ukraine; danilserg50@gmail.com; 5Pathology Department, Sumy Regional Oncological Hospital, 40022 Sumy, Ukraine

**Keywords:** hydroxyapatite, calcification, ovarian cancer, early diagnostics

## Abstract

Calcification is one of the clinical and morphological manifestations of ovarian tumors and it begins at the initial stages of carcinogenesis. Thus, this process can be used for the early diagnostics of some malignant ovarian tumors. We compared the results of ultrasound and histology and found that calcifications of a size less than 200 μm are not detected by ultrasound. These calcified structures are round fragile particles of different sizes. In the EDX (Energy-dispersive X-ray spectroscopy) spectra, the main lines were from Ca and P, and the ratio of these elements corresponds to hydroxyapatite. Thus, we established that hydroxyapatite is the main mineral component of ovarian psammoma bodies and could be used for early diagnostics of ovarian malignant neoplasia.

## 1. Introduction

Ovarian cancer (OC) is one of the leading causes of death among women worldwide [1]. Ovarian neoplasia ranks fourth among malignant tumors of the reproductive system in women after breast, uterus body and cervix cancer. Despite the lower prevalence of ovarian tumors compared with breast cancer, the mortality rate of this pathology is three times higher [1,2]. It is related to the asymptomatic course and ineffectiveness of screening diagnosis methods.

Calcification or biomineralization is one of the clinical and morphological features of ovarian tumor manifestation [3,4,5]. Pathological biomineralization (PBM) is detected in about 8% of OC cases by computer tomography [6]. Calcification is more common for serous ovarian adenocarcinoma [7,8,9]. Histologically, the incidence of calcifications in low-grade and high-grade serous carcinomas is 100% and 50%, respectively [10,11]. However, despite the relatively high detection rate, their prognostic and diagnostic values are not yet fully understood.

PBM in ovarian neoplasms can be divided into calcification of the stroma, capsule, tumor parenchyma, and psammoma bodies [12,13]. Pathological biomineral deposits start to develop in the earliest stages of carcinogenesis as nanocrystalline objects [11,13].

The study of the microstructure and phase composition of these crystallites is required for a deeper understanding of the processes of calcification in ovarian tumors. It may be the basis for the introduction of new early diagnosis and/or treatment methods. The specific crystal-chemical features of ovarian tumor calcifications are also relevant and require more detailed study [14].

Pathological crystalline inclusions are represented by nanoparticles. In some cases, they are X-ray amorphous and this complicates the study of their structure [15].

Comparison of the electrogram (picture of electron diffraction) with micromorphological features of crystal particles will allow more detailed study of keynote features of specific calcification types.

The aim of our work is to establish the morphological features and phase composition of pathological minerals and determine the potential diagnostic value of OC nanocrystallites.

## 2. Materials and Methods

### 2.1. The Ethics Committee

The study was approved by the ethics committee of the Medical Institute of Sumy State University (Proceedings 2/7, 14 July 2021).

For this study, we used 30 cases of serous ovarian carcinoma with PBM (group 1). Thirty cases of serous ovarian carcinoma without biomineralization (group 2) were used as a control group. All tissue samples were presented with surgical material after ovaryectomy and pangisteroectomy. Patients were operated on at the Sumy Regional Clinical Oncology Hospital (Sumy, Ukraine). Informed consent was obtained from each patient prior to admission to the clinic.

### 2.2. Ultrasound

We used Toshiba Applio MX with a linear multifrequency sensor 6–12 MHz (Tokyo, Japan) for ultrasound imagining at the private clinic “Floris” (Sumy, Ukraine).

#### Detection of Pathological Biominerals

Calcifications with a diameter of more than 0.05 cm were detected during grossing. The mineral component of macroscopic calcifications was separated from soft tissues by heat treatment at 200 °C for 1 h. This contributed to the destruction of the organic compounds of the calcification and the removal of water residues while maintaining the structure of the crystallite. Pathological biomineral formations with a diameter of less than 0.05 cm were detected by histology and scanning electron microscopy from histological sections of tumor tissue [16].

### 2.3. Histology

Ovarian tumor tissue was fixed in a neutral (buffered) 4% formaldehyde solution for 24 h, dehydrated and saturated with paraffin. Paraffin blocks were sectioned with a thickness of 4 μm with a Shandon Finesse 325 rotary microtome (Thermo Scientific, Waltham, MA, USA). After deparaffinization and dehydration (with xylene and ethanol), histological sections were stained with hematoxylin and eosin. All photos were captured with a digital visualization system based on a Zeiss Primo Star microscope with a Zeiss Axiocam ERc 5s digital camera and software package “Zen 2.0” (Carl Zeiss, Jena, Germany).

### 2.4. Histochemistry

To detect calcium deposition, we used von Kossa staining. Histological dehydrated sections of tumor tissue were treated with 5% aqueous solution of silver nitrate under the direct light of a 60 W lamp for 60 min followed by washing in sodium thiosulfate (5% aqueous solution). Nuclei were counterstained with an aqueous solution of nuclear fast red for 5 min (1:1000).

### 2.5. Immunohistochemistry (IHC)

Dehydrated sections were subjected to thermal unmasking of antigen in 0.1 M citrate buffer (pH 6.0) at 95–98 °C (Thermo Scientific, USA). We used UltraVision Quanto Detection System HRP and the DAB Quanto Detection System (Thermo Scientific, Waltham, MA, USA) for immunostaining and visualization. Sections were probed with anti-osteopontin (OPN) antibodies with dilution 1:200 (clone EPR21139-316, Abcam, Cambridge, UK). Nuclei were counterstained with Mayer’s hematoxylin. We used active (tissues with previously estimated positive and negative reactions) and passive (internal) control of IHC.

### 2.6. Scanning Electron Microscopy (SEM) with EDX

Histological sections with a thickness of 7 μm were mounted on a spectrally pure graphite base. Sections were preheated at 60 °C for 30 min, deparaffinized and dehydrated with xylene and ethanol. We used an SEO-SEM Inspect S50-B (SEO, Sumy, Ukraine) scanning microscope with an AZtecOne energy dispersion spectrometer with an X-MaxN20 detector (Oxford Instruments plc, Abingdon, UK). EDX spectra were analyzed with standard software of the microanalysis system.

### 2.7. X-ray Diffraction

Diffractometer DRON4-07 (“Burevestnik”, St. Petersburg, Russia) was used for X-ray diffraction of biominerals. Data were analyzed with the software package DIFWIN-1 (Etalon-PTC, Moscow, Russia). The phase composition identification was performed with the JCPDS database (Join Committee on Powder Diffraction Standards).

### 2.8. Transmission Electron Microscopy

A PEM-125K microscope (SELMI, Sumy, Ukraine) was used for transmission electron microscopy (TEM) with electron diffraction (ED). The powder of mineralized tissue was sonicated in distilled water with the UZDN-A sonicator (SELMI, Sumy, Ukraine). The specific power of the installation was 15–20 W/cm^2^ at a frequency of 22 kHz. The suspension (a few drops) was applied to the vertically upward ultrasonic emitter UZDN-A and sprayed for 2–3 s at optimal power. The sprayed aerosol was attached to a thin carbon film (10–20 nm) mounted on a copper mesh of the sample holder. ED pictures and microphotographs were captured at voltage U_(acceleration)_ = 90 kV.

### 2.9. Statistics

The normality of data distribution was checked by a Shapiro–Wilk test. Student’s *t*-test was applied for analysis of data with a normal distribution. Mann–Whitney’s U test was applied for nonparametric datasets. The results were considered statistically significant with a probability of more than 95% (*p* < 0.05). The correlation of parameters was tested with Spearman’s rank correlation coefficient. Statistical analysis was performed in Microsoft Office Excel 2016 with the addon AtteStat (version 12.0.5). All graphs were built in GraphPad Prism 7.

## 3. Results

### 3.1. Ultrasound

Ultrasound of ovaries with malignant tumors revealed solitary round hyperechogenic structures with smooth edges and clear contours, homogeneous echostructure and with no acoustic shadows. These formations ranged in size from 2 to 5 mm and had the avascular type of vascular pattern in the surrounding tissues (Figure 1A,B).

Clinical, macroscopic and ultrasound data are summarized in Table 1.

### 3.2. Histology

The OC tissue had micropapillary and macropapillary growths, single cells and chaotically formed small nests of cells with stroma infiltration (Figure 2A). Tumor cells were monomorphic, small, had moderate atypia of the nuclei and a defined nucleolus. Some samples of serous carcinomas were represented by solid masses with slit-like lumens, and tumor cells had large, hyperchromic, pleomorphic nuclei with chimeric configuration. Multinucleate cells with eosinophilic nucleoli were also found in tumor tissue. We found no necrotic changes in most of the serous carcinoma samples.

We detected the presence of PBM as psammoma bodies (PBs) in all samples. They were represented by concentric calcified structures and their fragments had an irregular shape. PB fragments formed due to mechanical damage by a microtome knife during the sectioning. Dot-like mineral deposits with polymorphic structure were also detected. The size of these formations ranged from 10 to 300 μm.

### 3.3. Histochemistry

Von Kossa’s staining revealed the presence of calcium phosphate. PBs and their fragments in the tumor tissue had a black and dark brown color. We noted the difference in the staining intensity between the core and outer layers of PBs (Figure 2B).

### 3.4. Immunohistochemistry

We revealed the OPN accumulation on the surface of biomineral formations by immunostaining. In general, OPN covered the surface of calcifications and was accumulated at the edges and between the lamellae of PBs. There was also positive cytoplasmic staining in tumor microenvironment cells, mainly in cells with mononuclear and fibroblast-like morphology (Figure 2C).

### 3.5. Scanning Electron Microscopy

SEM detected that calcifications were represented as round particles with a fragile structure and different sizes. It was confirmed by the presence of fragments of chimeric configuration. At high-power magnification, the fracture surfaces had a porous structure (Figure 2D). Nanocrystalline structures of spherical and needle shapes were found on the surface of calcifications.

In the EDX spectra, there were basic lines of Ca and P, as well as lines of O, C, Mg, Na and others. The ratio of the intensity of Ca and P lines corresponded to the hydroxyapatite Ca_10_(PO_4_)_6_(OH)_2_. However, there was a slight difference in the trace element composition, shown on spectra (Figure 2E,F).

### 3.6. EDX Mapping

According to maps of elemental distribution (Figure 3), we detected an increased concentration of calcium and phosphorus in the localization of calcified particles. Oxygen also was accumulated along with calcium and phosphorus. There was a fairly uniform distribution of carbon with no correlation with the localization of mineralized particles (calcification shielding of the carbon-containing surface of the graphite table from the electron beam).

### 3.7. TEM

TEM detected that apatite crystals were mostly polydisperse, but could also be monodisperse. The polycrystalline material of samples was identified by the ED pattern (Figure 4A).

The electron microscopic image showed relatively large crystals (40–50 nm) surrounded by small crystal particles (5–15 nm). It confirmed the polydisperse morphology of nanocrystals of pathological deposits (Figure 4B–D).

X-ray diffraction revealed that all pathological minerals were represented by calcium apatite with different levels of crystallinity (Figure 5). Similar results were previously published for biominerals of the gallbladder, prostate and thyroid gland [16,17].

## 4. Discussion

Calcification is one of the clinical and morphological features of ovarian tumor manifestation but it is not fully understood and needs to be studied in detail [18]. In the case of malignant ovarian tumors, the PBM begins at the earliest stages. It is caused by the partial death of malignant tumor cells. Their detritus is the basis for microcalcification development [19]. Thus, this process can be used for the early diagnostics of some malignant ovarian neoplasms.

Today, a number of routine and high-tech minimally invasive diagnostic methods are used, such as ultrasound, computed tomography (CT), magnetic resonance imaging (MRI), intravascular ultrasound (IVUS), optical coherence tomography (OCT) and positron emission tomography (PET). Technological advances make it possible to diagnose ever-smaller objects. However, these methods have different resolutions (Figure 6).

In the comparison of results of the ultrasound and histology, we noted that calcifications less than 200 μm in diameter were not detected by ultrasound. At the same time, CT scans detect pathological deposits of 1000 μm and bigger. IVUS visualizes calcification with a size of more than 100 μm. The newest high-tech and informative method is OCT. With this method, it is possible to recognize biomineral inclusions with a size of 10–20 μm and more. Shioi et al. reported that some calcifications may be X-ray amorphous [20]. This obviously depends on the composition of pathological biominerals, as well as on the maturity of the biomineral and its structure.

TEM and ED revealed the structural and morphological features of ovarian apatite crystals, which were not detected by other methods. Therefore, the development of a diagnostic method based on TEM with ED (or other high-resolution technique) in the future could improve the early diagnostics of malignant tumors with biomineralization.

We found that all biomineral deposits had calcium phosphate origin. The Ca/P ratio corresponded to hydroxyapatite. Hydroxyapatite has a direct effect on tumor activity and behavior. It also has an inhibitory effect on tumor proliferation and induces apoptosis [21].

IHC revealed OPN accumulation on the surface of biomineral deposits. OPN has an important role in the development and formation of bone tissue and PBM [22]. It binds to the surface of calcium phosphate crystallites (e.g., hydroxyapatite) and by so doing inhibits the calcification by limiting the biomineral growth [23]. Accordingly, OPN is detected at the sites of localization of calcified particles and is an absolute and early marker of PBM and the presence of hydroxyapatite.

Therefore, the study of the structure and physical, chemical and phase composition of OC calcifications, as well as features of their visualization, is important since provides the possible practical application of this pathological phenomenon for early diagnostics of OC and other neoplasms with biomineralization.

There was no significant difference in age between two groups of patients (*p* > 0.05). We found that tumors without PBM had significantly bigger sizes (*p* < 0.0001) (Figure 7). This indicates a possible positive role (tumor suppression) of calcifications in ovarian tumors. We also found a strong correlation between tumor size and calcification (r = 0.87, *p* < 0.05). Thus, the detection of nanocrystalline particles will contribute to the detection of tumors at the initial stages of cancerogenesis. This confirms our hypothesis about the possible application of high-resolution techniques for early cancer diagnostics.

## 5. Conclusions

We reported that ovarian cancer biominerals are represented by calcium apatite. No signs of other crystalline phases were detected.

The application of TEM with ED is perspective in terms of the effectiveness of high-resolution methods for cancer diagnostics by detection of pathological biominerals. According to TEM, pathological crystalline nanoparticles are polydisperse and their size ranges from 5 to 50 nm.

Thus, hydroxyapatite is the main mineral that is formed during the pathological biomineralization of ovarian malignancies. It could be used for early diagnostics of neoplasms of ovaries and other organs.

The identified relationship between tumor size and the presence or absence of calcifications is of interest for further study of the effect of hydroxyapatite compounds on tumor biology. Additionally, this fact can be used as a prognostic factor in oncology after large-scale clinical trials.

## Figures and Tables

**Figure 1 diagnostics-11-01741-f001:**
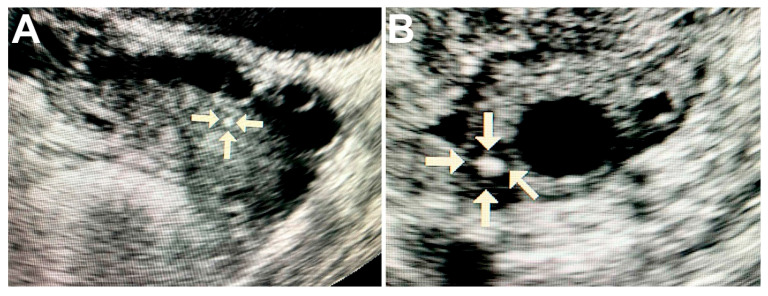
Ultrasound image of the ovary with pathological calcifications (marked by arrows). **A**—rounded biomineral structure in ovarian cancer tissue; **B**—the calcification is surrounded by cancerous tissue with cyst formation.

**Figure 2 diagnostics-11-01741-f002:**
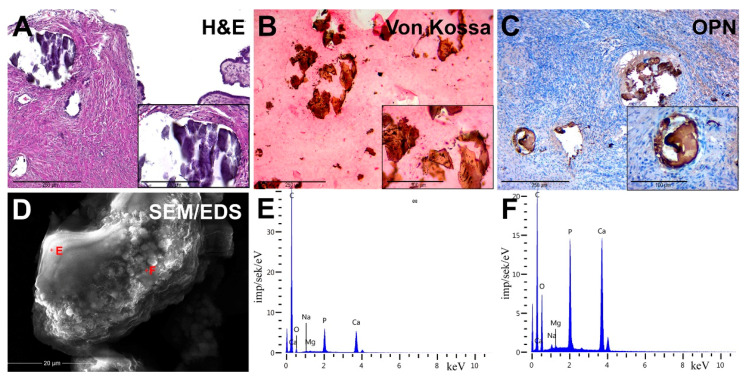
Pathological OC biominerals. (**A**) staining with hematoxylin–eosin; (**B**) von Kossa staining; (**C**) IHC detection of osteopontin (OPN); (**D**) Scanning Electron Microscopy (SEM) with energy-dispersive X-ray spectroscopy (EDX), analysis points are marked by red crosses; (**E**,**F**) EDX spectra of ovarian cancer calcifications.

**Figure 3 diagnostics-11-01741-f003:**
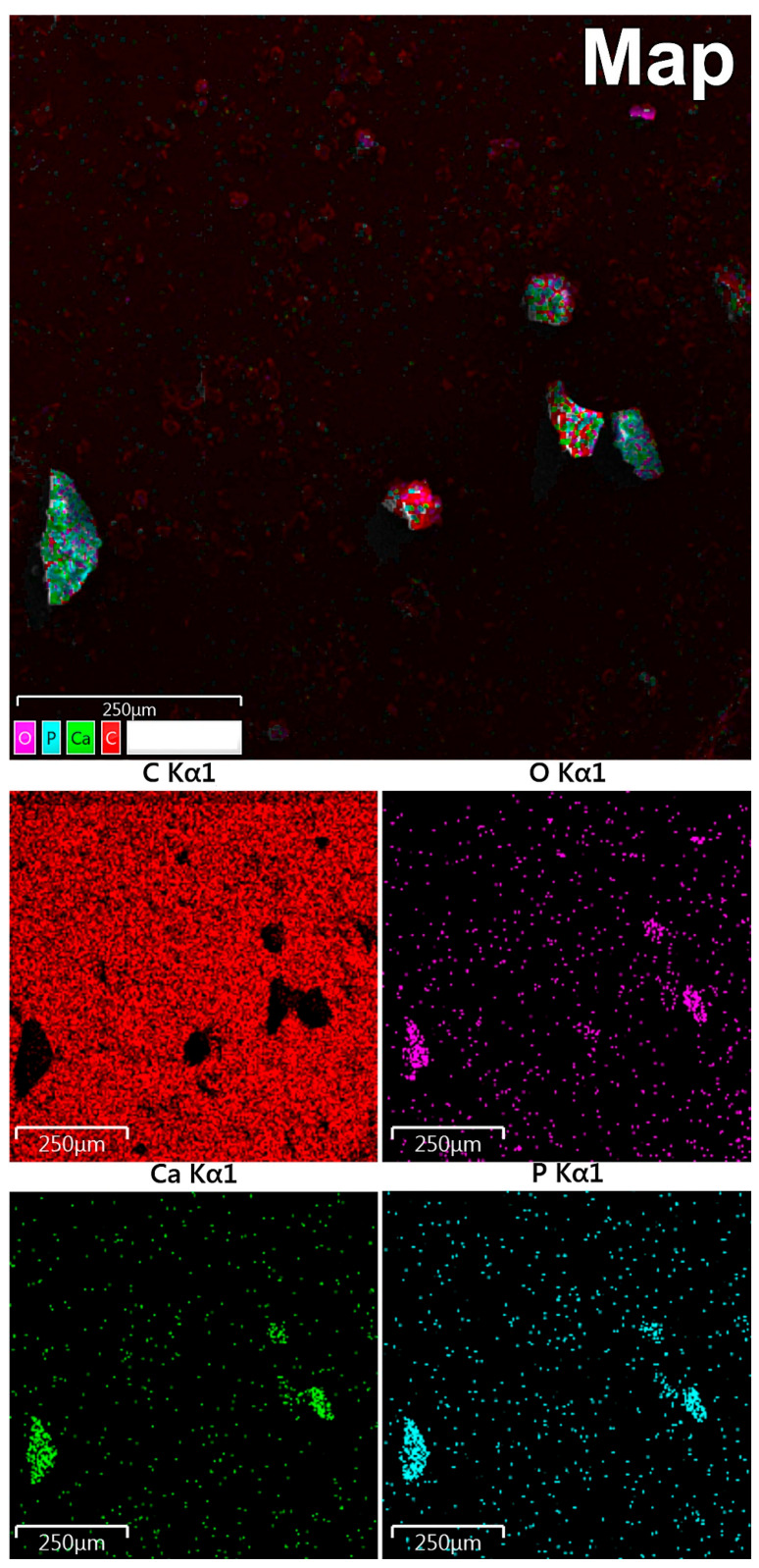
Study of OC calcification by EDX mapping: red indicates carbon, purple—oxygen, green—calcium, blue—phosphorus.

**Figure 4 diagnostics-11-01741-f004:**
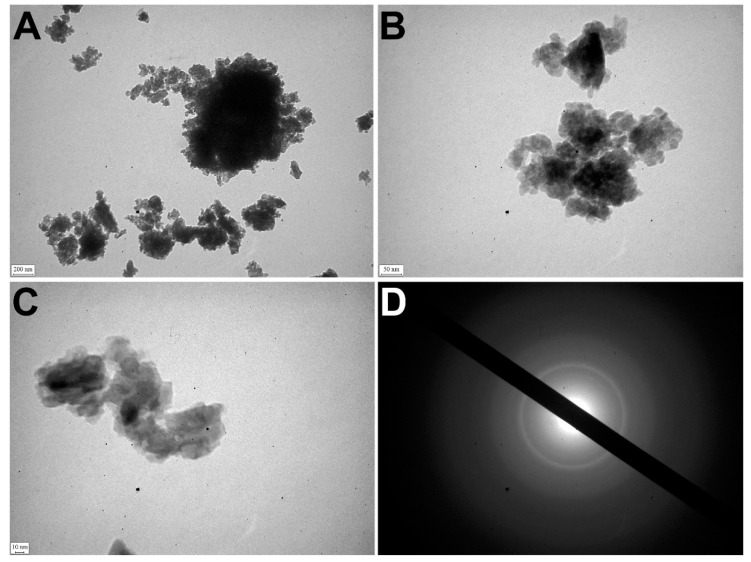
TEM of OC calcified samples. (**A**–**C**) TEM images of nanocrystals; (**D**) ED image.

**Figure 5 diagnostics-11-01741-f005:**
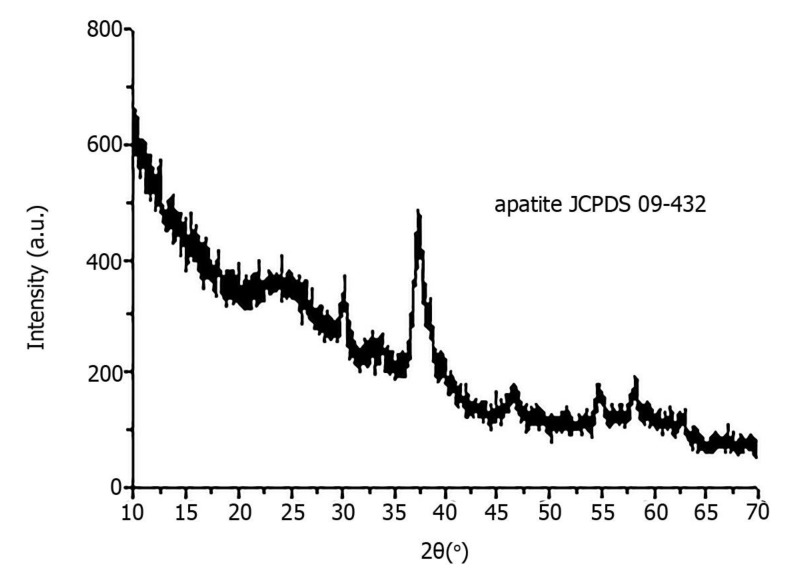
X-ray diffraction of pathological biomineral of ovarian tumor (sample 23).

**Figure 6 diagnostics-11-01741-f006:**
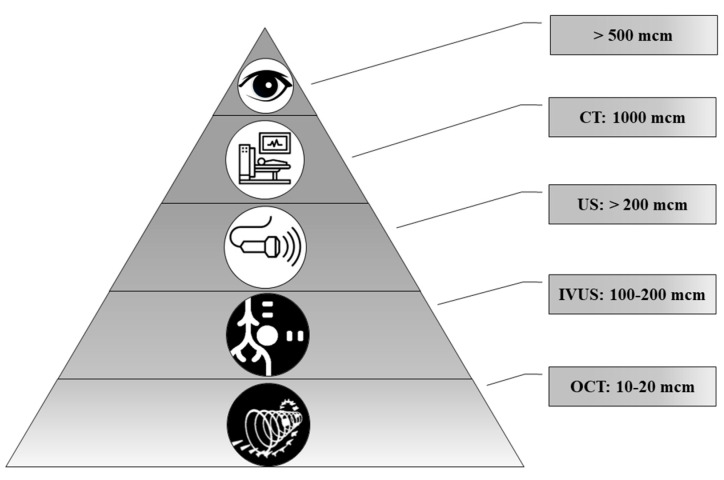
The representation of resolution of diagnostic methods used for biomineral detection.

**Figure 7 diagnostics-11-01741-f007:**
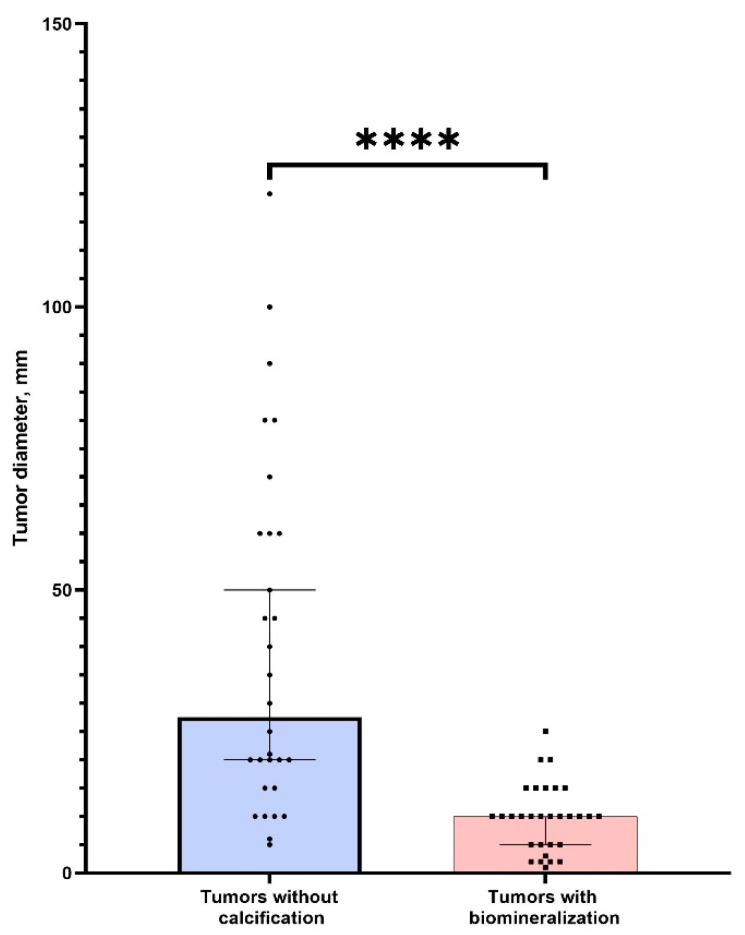
The comparison of tumor sizes (larger diameter) in groups of samples with the presence and absence of calcification. **** corresponds (*p* < 0.0001).

**Table 1 diagnostics-11-01741-t001:** Ovarian tumors with calcifications.

Case	Age of Women	TNM			Tumor Diameter (Larger Size), mm	Diameter of Calcifications by Ultrasound (Larger Size), mm
		T	N	M		
**1**	44	T1	N0	M0	2	0.5
**2**	59	T2	N0	M0	10	1.0
**3**	30	T1	N0	M0	1	0.25
**4**	59	T2	N0	M0	10	1.0
**5**	57	T2	N0	M0	10	2.0
**6**	44	T3	N1	M0	15	2.0
**7**	57	T1	N0	M0	5	1.0
**8**	65	T1	N0	M0	3	0.5
**9**	48	T3	N1	M1	15	2.0
**10**	37	T2	N0	M0	10	0.8
**11**	58	T1	N0	M0	5	1.0
**12**	52	T2	Nx	M0	10	1.0
**13**	77	T1	N0	M0	2	0.5
**14**	51	T2	N0	M0	10	1.0
**15**	56	T1	N0	M0	2	0.5
**16**	50	T2	Nx	M0	10	1.0
**17**	58	T3	N1	M1	20	3.0
**18**	46	T2	N0	M0	10	1.0
**19**	55	T2	N0	M0	10	1.0
**20**	43	T2	N0	M0	15	2.0
**21**	69	T3	N1	M1	20	4.0
**22**	66	T1	N0	M0	5	1.0
**23**	57	T2	N0	M0	10	2.0
**24**	53	T4	N1	M1	25	4.0
**25**	72	T3	N1	M0	15	3.0
**26**	64	T1	N0	M0	5	1.0
**27**	50	T2	N0	M0	10	2.0
**28**	54	T1	N0	M0	2	1.0
**29**	63	T3	N1	M0	15	2.0
**30**	57	T2	N0	M0	10	1.0

Notes: T—tumor size, N—metastases in lymphatic nodules, M—distant metastases. The data table of the control group is presented in the Appendix A.

## Data Availability

Data available within the article or its Appendix A.

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
