# Peer review of "Potential Role of Hydroxyapatite Nanocrystalline for Early Diagnostics of Ovarian Cancer"

_diagnostics, 2021, doi:10.3390/diagnostics11101741_

Round 1

Reviewer 1 Report

Thank you for asking me to review this interesting paper.

In general the paper has a good value for publication however I have these 2 comments

1- the methods: Please describe when the scan was performed, in vivo or in vitro as it is not clear how the scan was done. 

2- Was informed consent done

3- Was the scan TV in all patiebts

4- what were the charcteristics of the tumors: size, shape, stage, all this need to be revealed

Results: There is no numbers or stats in the results. It will be very useful to see tables of results illustrating numbers of cases with different diameters and detection (yes/no) by ultrasound. 

Author Response

Dear Reviewer!

We are very grateful for the opportunity to publish our paper in your journal.

We have considered all revivers’ comments and believe that they will improve this paper.

1.     Scanning electronic microscopy was performed on tissue samples. We made these samples from histological paraffin blocks of tissues (method description in Moskalenko R. et al., Acta Facultatis Medicae Naissensis, 2020).

2.     We have obtained informed consent from all patients during the hospitalization to the hospital. This research was also discussed and approved by the Bioethics Committee of the Medical Institute (protocol № 2/07).

3.     All patients were examined by ultrasound with Toshiba Applio. The technical features and capabilities of this device at the local private clinic Floris (http://www.floris-sumy.com.ua) determine the use of TV scans as the best technique to illustrate our study.

4.     We included a table with examination results (such as the size and characteristics of ovarian tumors and their calcifications) in the paper. We also added a control group of 30 samples without calcification. These data will be published in the section "Research Results".

Reviewer 2 Report

This paper presents the potential promise of analysing calcified structures for the early detection of ovarian cancer. While the message is of extreme importance, some issues raised below should be further addressed in order to achieve a clearer message:

  • current design only includes 30 cases of serous ovarian carcinoma. There was no control group included. Is it possible to find such calcified structures in women without ovarian cancer? What are other possible conditions where such structures may appear?
  • The message of the manuscript aims at early detection. However, no information is provided on the stage of cancer cases. For the early detection it is crucial that manifestation starts early enough. Would authors please include the table with demographic characteristics of patients together with cancer stage.
  • in the Results section authors present different ranges of the sizes of formations found by different methods. Section would surely benefit from some table that would summarize the distributions of these sizes (mean with s.d. or median with IQR). It is also interesting if sizes correlate with any of demographics or stage

Author Response

Dear Reviewer!

We are very grateful for the opportunity to publish our paper in your journal.

We have considered all comments and believe that they will improve this paper.

  1. We included data of the control group (30 patients) in the section "Research results," such as age, tumor size, stage of the tumor process, and the size of calcifications according to the ultrasound.
  2. Calcifications appear in the ovarian tissue only in the case of pathology. An important factor in the formation of calcifications is necrosis, which most often appears during the death of tumor cells [Wen J. et al., J Cancer, 2018]. A special form of ovarian calcifications is psammoma bodies, which have a specific round lamellar shape and are a specific sign of a malignant tumor - serous papillary ovarian carcinoma [Rosen D. et al., J Biomed Res, 2010; Das D.K. Diagn Cytopathol, 2009]. Our work is devoted to the study of serous papillary ovarian carcinoma.
  3. We added general information about the size and stages of tumors, the size of calcifications, and the correlation between them.
